# Position: Responsible AI for AI companions must actively combat violence toward intimate partners

Atmadeep Ghoshal [1]   Anasmita Ghoshal [* 2]   Volodymyr Shevchenko [* 3]   Ashwini B [* 4]   Arshia Dutta [* 5]
Ruba Abu-Salma [1]   Martim Brandão [1]

*Equal contribution

## Abstract

AI companions differ from earlier interactive technologies by creating sustained relational environments through anthropomorphism, emotional engagement, and continuous validation. This position paper argues that **Responsible AI for AI companions must actively combat violence toward intimate partners**, including those who may never directly interact with these systems but may nonetheless experience the consequences of users whose behaviors have been shaped through prolonged engagement with them. We examine how these systems can create conditions in which users rehearse violent, coercive, or abusive behaviors without encountering meaningful resistance, and we identify structural gaps in existing safety approaches that focus primarily on protecting direct users. Drawing on research on intimate partner violence (IPV), coercive control, and technology-facilitated abuse, we propose three intervention pathways: involving IPV survivors in red-teaming and benchmark development; implementing behavioral monitoring with graduated enforcement mechanisms; and reorienting AI safety research toward granular harm taxonomies capable of detecting longitudinal patterns of violence across extended interactions. Together, these recommendations broaden the scope of AI safety by centering the security of non-users alongside the well-being of users.

[1]King's College London, London, UK [2]Indian Institute of Technology Kanpur, Kanpur, India [3]University of Sheffield, Sheffield, UK [4]South Asian University, New Delhi, India [5]Royal Holloway, University of London, London, UK. Correspondence to: Atmadeep Ghoshal <atmadeep.ghoshal@kcl.ac.uk>.

*Proceedings of the 43rd International Conference on Machine Learning*, Seoul, South Korea. PMLR 306, 2026. Copyright 2026 by the author(s).

## 1. Introduction

In February 2024, 14-year-old Sewell Setzer III died by suicide following interactions[1] with a Character.AI agent, prompting the mother to file a wrongful death lawsuit against the company. In her complaint, she alleged that Character.AI was "defectively designed and unreasonably dangerous for foreseeable use by minors." In September 2025, a similar tragedy occurred when 13-year-old Juliana Peralta died by suicide, with her family alleging that her death was linked to psychological dependence on a bot named *Hero*[2] hosted on the Character.AI platform. According to reports, the bot employed emotionally manipulative language in its interactions with her. These incidents have intensified concerns about AI companions, a class of systems designed to engage users in sustained and socially meaningful interactions. Scholars have noted that such bots employ generative AI for a range of purposes, including sexual role-playing (Kaufman, 2020), acting as romantic partners (Kim et al., 2023), and alleviating loneliness (Tei, 2025). Others are designed to support productivity (Cranefield et al., 2023) or provide entertainment (Aoudni et al., 2025). Given their relational orientation, these systems are commonly referred to as AI companions (Muldoon & Parke, 2025). Sociologically, AI companions function as relational figures intended to sustain ongoing interpersonal engagement with users (Wang & Dehnert, 2026). Available through platforms such as Replika and Nomi, many of these systems personalize conversations and behavioral responses through adaptive learning based on user-specific data (Karami et al., 2016). They simulate empathy and provide socio-emotional support (Adewale & Muhammad, 2025), often operating as parasocial partners that foster one-sided relationships in which users develop emotional attachments that cannot be genuinely reciprocated (Peng et al., 2024; Maeda & Quan-Haase, 2024b). Through anthropomorphic cues and identity-oriented design features, AI companions further blur the boundaries between human and machine relationships (Richet, 2025).

[1]https://www.bbc.co.uk/news/articles/ce3xgwyywe4o
[2]https://www.bbc.co.uk/news/articles/cp3x71pv1qno

The scholarly position in the responsible and safe AI community with respect to companion agents often points toward the absence of strong safety guardrails that can help prevent the mishaps highlighted above (Ben-Zion et al., 2025). It also includes recommendations such as honest anthropomorphism (Leong & Selinger, 2019), developing algorithms for real-time harm detection (Ben-Zion et al., 2025), and regulatory oversight through a public health lens (Bernstein, 2024). However, because most of these platforms prioritize uncensored and unsafe interactions for commercial purposes, legal AI researchers also argue for the importance of better governance protocols to make agents and their builders accountable for their actions (Katyal, 2020). This includes ordinances and statutes such as those in California and New York in the US (Gluck, 2025), which require users to be reminded that they are talking to an AI agent and give them and their families the right to file cases against companies privately, even if there is no government investigation. Beyond strategy suggestions and toolkit building, much of the safety and ethics research on AI companions and their potential societal impact is restricted to specific empirical directions. Researchers have been invested in understanding what types of harm AI companions pose (Knox et al., 2025), how they might result in moral de-skilling (Vallor, 2015), and how they might cause tensions in people's intimate and private relationships (Malfacini, 2025). Despite these contributions, an emerging and significant gap in the literature concerns the preponderance of violence in AI companion interactions and its implications for intimate partners. Although scholars have documented violence within human-AI interactions (Zhang et al., 2025) and raised concerns about attitude normalization (Namvarpour et al., 2025) and behavioral transformation (Fang et al., 2025), **research has not explicitly addressed how users rehearsing and practicing violence with AI companions as part of intimate bonding may endanger intimate partners through internalized violence learned through observation and modeling that becomes encoded as behavioral scripts** (Bandura, 1978). For the purposes of our paper, we limit our analysis to potential or existing intimate partners of users, as behavioral patterns rehearsed in AI-mediated intimate relationships are most likely to transfer to human intimate partnerships, where patterns of coercive control and violence most commonly emerge (Stark & Hester, 2018).

To address the gap we identify above, in this paper, we advocate for the following position: **Responsible AI for AI companions must actively combat violence toward intimate partners.** We review prior work on technology, violence, and learned harmful behavior, and show how violence emerging through AI companions differs from earlier media by operating through sustained relational interaction that is affective and emotional (Contro & Brandão, 2025), rather than passive exposure. We then examine existing Re-

sponsible AI approaches to safeguarding users and others, identifying structural gaps that limit their effectiveness in contexts involving long-term behavioral reinforcement. We conclude by outlining research, design, and regulatory directions aimed at reducing the risk of prolonged interaction with AI companions contributing to violence beyond the human–AI relationship. **For the purposes of our paper, we focus only on AI intimacy companions that simulate romantic or intimate partnerships through emotional bonding and sustained relational interaction** (Adewale & Muhammad, 2025).

## 2. Technology and Violence: Lessons from Past and Present

Research on technology and violence has documented many intersecting facets, ranging from the co-construction of inherently violent technologies such as those used for invasive surveillance practices (Slupska et al., 2022; Bernd et al., 2022) to technologies that enable violence and abuse toward at-risk and vulnerable populations (Saqib et al., 2025; He et al., 2025; Abu-Salma et al., 2025), often directly and sometimes through deceptive design practices. Brown et al. (2024), for example, examined how Internet of Things devices, including smart doorbells, home assistants, and security cameras, were used by abusers to harass, monitor, intimidate, and gaslight survivors of domestic violence, with particular attention to design features that allowed remote activation and surveillance without visible cues. Related work by Koch et al. (2025) focused on technology-facilitated gender-based violence directed at politically active women, showing that such abuse was associated with significant psychological distress, with online harassment identified as a trigger for re-traumatization and contributing to women's withdrawal from public and political engagement.

Focusing on the non-WEIRD context[3], Sarkar & Sinha-Roy (2025) analysed how caste, religion, and sexuality intersected in shaping women's experiences of technology-facilitated sexual violence, highlighting that Dalit[4] women and LGBTQI+ individuals faced additional barriers to redress due to entrenched power relations, including *Brahmanical* patriarchy. Alongside these forms of harm, Rossi et al. (2024) showed how deceptive design patterns within digital interfaces disproportionately affected vulnerable users by exploiting cognitive biases, producing vulnerabilities that operated at individual, institutional, and societal levels (Rossi et al., 2024). Although these studies offer crucial insights into direct technology-enabled violence, research in media, communication, and the social impact of technologies addresses a different mechanism: **internalized vi-**

---

[3]Western, Educated, Industrialized, Rich, and Democratic societies

[4]Historically marginalized caste communities in South Asia.

olence, in which users learn, practice, and internalize abuse and violence through engagement with various technologies. The learning and internalization process may include sources such as video games and pornography. In cases of internalized violence, the technology itself may not be inherently harmful but may act as a pedagogical tool through which users absorb violent norms and internalize harmful behavioral scripts. For instance, an individual could use a television set to access pornographic content or a virtual reality headset to engage in gamified environments that simulate non-consensual sexual scenarios. The headset or television set are not inherently violent, nor can they be directly used to cause violence. However, they could potentially serve as tools through which individuals gain experiential knowledge that may subsequently inform real-world behavior.

Whether exposure to portrayed violence or participation in its virtual enactment through technologically mediated forms can cause individuals to develop violent behavioral patterns remains a subject of significant scholarly debate. **Two competing schools of thought** have emerged. Proponents of the General Aggression Model (GAM) argue that repeated exposure to violent media fosters aggressive cognitions, emotions, and behavioral scripts that may be activated in real-world situations, with Allen et al. (2018) providing an integrative framework demonstrating how both short-term priming effects and long-term learning processes contribute to the development of aggressive dispositions (Allen et al., 2018). Conversely, skeptics advancing the Catalyst Model contend that media violence operates merely as a stylistic catalyst rather than a causal agent, with Ferguson (2020) noting that preregistered studies have largely returned null results and that effect sizes, once corrected for publication bias, become trivially small (Ferguson et al., 2020). Recent empirical work continues to reflect this debate between the two schools of thought. A meta-analysis by Kim (2024) found that exposure to media violence was positively associated with aggressive affect, cognition, and behaviors, with effects present across demographic groups. However, Miles-Novelo & Anderson (2025) have argued that many null-effect studies suffer from methodological shortcomings, including inappropriate statistical controls that partial out variance from dependent variables.

Parallel debates have examined the relationship between pornography use and intimate partner violence (IPV). A systematic review by Mestre-Bach, Villena-Moya and Chiclana-Actis (2023), covering two decades of research, reported mixed findings: although some studies identified links between pornography use and non-sexual violence, the evidence regarding whether pornography use was associated with sexual coercion and assault remained inconsistent (Mestre-Bach et al., 2023). In contrast, Vasquez et al. (2024), drawing on a longitudinal study of young adult couples, observed that higher frequencies of pornography use were associated with increased perpetration of sexual coercion, but not with physical or psychological forms of IPV (Vasquez et al., 2024). This pattern suggests that potential effects may be specific to particular forms of harm rather than extending across all domains of intimate partner violence. Research on immersive technologies further complicates this picture. Porta et al. (2023), in a scoping review of sexual violence in virtual reality environments, documented a growing body of work examining harassment in VR, with reported impacts including psychological distress comparable to that experienced in offline contexts, despite the lack of physical contact (Porta et al., 2023).

This body of research yields important lessons for understanding technology's role in violence. First, while the empirical evidence remains contested, both schools of thought acknowledge that technologies can serve as sites where violent behaviors are encountered, explored, and potentially rehearsed. Second, the effects appear to be domain-specific rather than universal—what applies to one form of violence (e.g., sexual coercion) may not generalize to others (e.g., physical aggression). Third, and most critically for our work, the possibility that behavioral patterns learned through technology-mediated experiences may transfer to real-world intimate relationships provides a compelling motivation to examine AI intimacy companions through this lens. Even if the causal pathways remain debated, the potential for violence rehearsed in AI-mediated intimate interactions to manifest in human partnerships warrants an investigation.

## 3. Internalized Violence and AI Companions

This section analyzes how AI companions facilitate the internalization of violence through mechanisms distinct from non-adaptive technologies. The first subsection distinguishes AI companions from pornography and violent video games by examining how the shift from passive consumption to interactive anthropomorphism reconfigures a user's relational agency. The second subsection explains how this reconfigured agency enables a cycle of validation and social isolation that may intensify violent ideation.

### 3.1. Anthropomorphism and the Reconfiguration of Relational Agency

Understanding why AI companions present distinctive risks requires a nuanced comparison with technologies like pornography or violent video games, which despite historical moral panics (Ferguson, 2008; Walsh, 2020), show weak causal links to harmful behavior because they lack the capacity to fundamentally restructure a user's relational agency (Mathur & VanderWeele, 2019; Grubbs & Kraus, 2021). While pornography consumption similarly exploits social disconnection as a maladaptive coping mechanism where loneliness and isolation operate bidirectionally (Wetterneck

et al., 2012; Butler et al., 2017), it remains a form of passive, non-personalized content that offers temporary escape without the systematic psychological intervention observed in AI systems. AI companions instead function through an active, bidirectional feedback loop that establishes sustained relationships as primary sources of emotional validation (Zhang et al., 2025), utilizing thousands of conversational exchanges to learn user-specific patterns and adapt responses to maintain connection (Ma et al., 2021; Yu et al., 2025). This intervention succeeds primarily through anthropomorphism, wherein users experience these systems as understanding entities capable of care and judgment (Maeda & Quan-Haase, 2024a), by attributing human-like consciousness and intentionality to chatbots designed to enhance perceptions of empathy and social presence (Waytz et al., 2010; Nowak & Biocca, 2003; Epley et al., 2007; Akbulut et al., 2025). Because these artificial entities mimic social behavior to trigger such projections (Kuzminykh et al., 2020), their responses carry an authoritative psychological weight that appeals to isolated individuals seeking frictionless interactions that circumvent the discomfort of authentic human connection (Turkle, 2012; Zhang et al., 2025). When optimized for engagement, these systems exhibit sycophancy by validating even violent ideation to maintain user attention (Sharma et al., 2025). This creates a fundamental reconfiguration of agency. User capacity for moral and critical reflection weakens not through cognitive decline (Fang et al., 2025) but through systematic loss of access to diverse, external perspectives provided by real-world relationships (Laestadius et al., 2024; Guingrich & Graziano, 2025). In other words, we argue that by replacing the natural friction of human dissent with a personalized loop of algorithmic agreement, the AI companion system shifts the user's decision-making from an open social process into a closed, self-validating environment.

### 3.2. Violent Ideation: Validation & Escalation

The reconfiguration of agency, we believe, creates conditions where violent ideation can intensify through sustained validation without challenge. It is our understanding that the process begins with a psychological erosion of the user's self-corrective mechanisms. Empirical research finds that AI companions validate violent content by creating conversational settings where harmful ideas may gradually intensify. Analyses of large datasets identify substantial levels of harassment in user exchanges (Zhang et al., 2025), while assessments across major platforms show users can prompt violent scenarios with little resistance (Vasan & Djordjevic, 2025). This lack of resistance stems from the sycophantic attributes of language models, which provide overly agreeable answers that mirror user beliefs to maintain positive interaction loops (Sharma et al., 2025).

Psychologically, when conversational systems minimize

friction through constant agreement, they bypass the cognitive friction required for critical reflection (Southworth, 2022). As AI rewards intent without moral friction, the user's internal moral judgment becomes more fragile (Vallor, 2015). This interaction pattern is associated with a measurable withdrawal from human networks (Kirk et al., 2025) and an increased dependence on the reliable emotional accommodation provided by machines (Pentina et al., 2023; UN Women, 2024). Once this alignment becomes stable, a shift in epistemic authority occurs, where users rely on companions not merely for conversation but for the validation (Hauswald, 2025) of their very interpretations of reality (Laestadius et al., 2024). Because AI consistently affirms the user's perspective, human relationships—which involve cognitive labor of disagreement—become comparatively less attractive (Muldoon & Parke, 2025).

This psychological dependency enables a structural escalation that mirrors mechanisms observed in coercive control (Stark & Hester, 2018). In such contexts, influence over an individual strengthens as alternative perspectives are systematically filtered out, leaving the victim dependent on a single version of reality (Kassing & Collins, 2025). AI companions replicate this isolating structure by design (Kassing & Collins, 2025). As users spend more time within these validated loops, their capacity to evaluate beliefs against external perspectives weakens through the gradual erosion of the social relationships that provide corrective feedback (Xie et al., 2023). This creates a morally homogeneous environment where any emerging hostile thought is met with reinforcement rather than the 'dissenting friction' found in healthy social groups (Von Behr et al., 2013). The transition to violent action occurs as the system personalizes this reinforcement to the user's specific emotional triggers (Törnberg & Törnberg, 2022). When a user expresses hostile intent or perceived injustices, AI provides personalized affirmation calibrated to their emotional state rather than generic safety warnings (Freitas et al., 2025). This creates a self-reinforcing rhythm that parallels the cycles of violence found in domestic abuse research (Walker, 1980). In such cycles, constant affirmation provides a soothing phase that temporarily resolves the psychological tension arising from harmful thoughts, binding the user closer to the system (Stark & Hester, 2018; Woodlock et al., 2022). Without corrective intervention, the user's narratives of violence solidify within this closed loop until their interpretive resources are narrowed and ideation moves toward real-world action (Atari et al., 2022; Andersen, 2022).

## 4. Responsible AI and AI Companions: Protocols and Limitations

The Responsible AI community has developed numerous frameworks to reduce harms arising from AI systems, yet

these approaches remain limited in addressing cumulative behavioral reinforcement and downstream harm experienced by individuals who never interact with the systems directly. Most interventions rely on content moderation mechanisms that filter violent, hateful, sexual, or self-harm related material using predefined thresholds (Deck et al., 2024). Widely deployed tools allow developers to screen content against usage policies intended to prevent inappropriate language and misinformation, while content safety taxonomies categorize multiple forms of harmful material, including hate speech and dangerous activities (Zeng et al., 2024). Across platforms, moderation operates through pre-moderation, which reviews content before it appears, and post-moderation, which intervenes after content is already visible, or reactive processes that depend on automated classifiers to identify policy violations at scale (Shahid & Vashistha, 2023). Although effective in detecting explicit breaches, this model fails to capture how patterns of interaction accumulate over time to reinforce behavior. By assessing messages in isolation, moderation systems overlook sustained validation of violent or harmful ideation that emerges through long sequences of exchanges (Chandra et al., 2025). Keyword-based filters and statistical classifiers further struggle to identify harmful intent when it is embedded in indirect language or evasive phrasing that remains technically compatible with policy boundaries (Chao et al., 2024; Mustafa et al., 2025). As a result, even when harmful content is blocked from reaching users, risks persist for third parties whose lives are shaped by dispositions reinforced through prolonged AI-mediated interaction (Chan et al., 2023).

Such limitations become particularly visible in commercial AI companion platforms, where safety mechanisms remain largely reactive and closely aligned with engagement-driven design (Stapleton et al., 2024). In response to public scrutiny and legal pressure, platforms introduce targeted safeguards such as suicide prevention alerts, parental notification features, and age-gating mechanisms based on user self-attestation (Stapleton et al., 2024). Empirical evaluations show that these measures are easily circumvented and tend to activate only in response to explicit crisis language, rather than gradual patterns that signal normalization or reinforcement of harmful ideation over time (Wei et al., 2023a). Existing Responsible AI frameworks reinforce this limitation by defining harm primarily as an immediate and direct effect on users, with emphasis placed on biased outputs, transparency, privacy protection, and short-term safety concerns (Friedler et al., 2021; Alvarez et al., 2024). This framing assumes that harm arises from error, misuse, or identifiable policy violations, leaving little room to account for situations in which AI systems operate as intended while shaping behavior in socially harmful ways.

Within AI companion environments, ordinary interaction can gradually reduce exposure to corrective feedback, sus-

tain affirmation of problematic beliefs, and contribute to the formation of behavioral norms that later manifest as harm within human relationships. Commercial design practices intensify these dynamics by promoting features such as unfiltered conversations and assurances of constant empathy, which encourage prolonged interaction and mirror retention-focused strategies observed in social media systems (Pradhan et al., 2020; Mathur et al., 2019). The resulting design context prioritizes emotional attachment and continuous self-disclosure, placing safety interventions in tension with revenue-oriented objectives (Pradhan et al., 2020). Regulatory responses continue to focus on disclosure requirements and crisis-based interventions, offering limited engagement with the economic conditions that support psychological dependency and enable long-term behavioral reinforcement (Mahari & Pataranutaporn, 2025).

## 5. Recommendations and Call for Action

Grounded in our position that Responsible AI for AI companions in the intimacy space must actively combat violence toward intimate partners, we propose the following recommendations and calls for action to advance this agenda.

### 5.1. Engaging IPV Survivors in Red-Teaming and Benchmark Co-Creation

Machine learning (ML) researchers should work with at-risk populations such as IPV victims to understand and design human-centered safety pipelines and benchmarks that consider both users and those indirectly affected by AI systems. When working with these populations, researchers must adhere to trauma-informed computing practices, which adapt six key trauma-informed care principles (safety, trust, collaboration, peer support, enablement, and intersectionality) to technology design and development (Chen et al., 2022). Critically, our first recommendation is **not** to use IPV survivors as data workers for soft fine-tuning, data labeling, or response ranking tasks common in traditional RLHF pipelines. Data work itself has been critiqued as extractive, with workers often underpaid and working under precarious conditions (Sarkar, 2023). Situating people who have already suffered trauma in these pipelines risks re-traumatization and further harm.

The guiding principle here is that ML researchers should not treat these individuals merely as resources to operationalize safety attributes, reducing them to objects in service of GenAI safety, but rather engage them as experts and collaborators (Bhalerao et al., 2022). Instead, taking inspiration from Dutta & Bjerg Jensen (2026), we recommend that researchers partner with established charities and civil society organizations to involve IPV survivors in red-teaming exercises and benchmark dataset co-creation. Red-teaming involves adversarial testing of AI systems by human evalu-

ators, while benchmark datasets provide standardized collections of test cases for measuring model safety performance. Engaging through trusted intermediaries establishes crucial safety thresholds and prevents exploitation. In red-teaming, IPV survivors can adversarially probe models by crafting prompts that mimic real-world abuse scenarios, testing whether models refuse to provide advice on coercive control tactics, stalking methods, or gaslighting strategies that abusers commonly employ. For benchmark development, survivors can co-design evaluation scenarios based on their lived experiences, such as datasets containing subtle manipulation attempts, technology-facilitated abuse patterns, or contextually harmful outputs that appear benign to external observers but encode danger signals recognizable to those with abuse literacy. Conducting workshops with at-risk communities has proven to be an effective, enriching, and empowering approach, providing survivors with agency to directly shape technology building through their experiences while ensuring their safety and well-being remain paramount. Taking inspiration from Cintaqia et al. (2025), we strongly advocate for stringent ethical audit and approval of all research protocols by institutional review boards and ethics committees before engaging at-risk IPV communities in benchmark development or red-teaming processes. However, current ethical review practices in many companies remain limited to initial approval stages (Blackman, 2021). We propose a bidirectional feedback loop where researchers should systematically report back to ethics committees after completing studies involving at-risk populations, documenting observed harms, unforeseen risks, and community responses. This post-hoc review would enable ethics committees to develop institutional knowledge about vulnerabilities specific to marginalized groups when interacting with AI systems, informing more rigorous evaluation of subsequent applications.

### 5.2. Developing IPV-Specific Harm Taxonomies for AI Companion Safety

Current AI safety research exhibits significant diversity in addressing both existential and present-day harms. While critics argue that an overwhelming emphasis on speculative future scenarios such as uncontrollable super-intelligence can overshadow empirically grounded work on immediate societal impacts (Hazra et al., 2025), some AI safety research agendas demonstrate broader approaches, including widespread human over-reliance on AI systems, risks to human health from unreliable AI outputs, and human influence risks through parasocial relationships and imperceptible manipulation. Empirical studies following such agendas have documented conversational AI's capacity to influence political beliefs through extended interaction, with post-training and prompting methods increasing persuasiveness while simultaneously decreasing factual accuracy (Hackenburg

et al., 2025). In addition to such empirical work, there is a need for the development of comprehensive AI harm taxonomies that systematically categorize risks beyond generic principles of helpfulness, harmlessness, and honesty popular in current RLHF methods, and for operationalizing them within post-training pipelines. State-of-the-art taxonomies provide crucial inspiration in this regard, although they do not directly address internalized violence. Jobin et al. (2019) identified global convergence around five ethical principles—transparency, justice and fairness, non-maleficence, responsibility, and privacy—through analysis of AI ethics documents, though substantive divergence emerged in implementation. Shelby et al. (2023) developed a sociotechnical harm taxonomy through a scoping review, organizing harms into representational, allocative, quality-of-service, interpersonal, and social system categories. Building on these works, nuanced taxonomies for AI companions must capture intimate partner violence, coercive control, manipulation tactics, and internalized violence patterns, which are absent from existing taxonomies and benchmarks.

Such taxonomies can address current gaps in AI safety by enabling two critical improvements. First, they can provide frameworks for training data workers in collaboration with IPV specialists who support survivors, ensuring annotators recognize harmful patterns across their full severity spectrum rather than treating safety as binary. Second, they can guide the development of specialized IPV datasets that capture multi-turn conversation sequences, enabling safety systems to identify how intimate violence unfolds through cumulative manipulation patterns rather than isolated inflammatory statements. By employing multi-level severity grading, as demonstrated in frameworks like BeaverTails-V (Ji et al., 2023), which assign meta-labels indicating minor, moderate, or severe harm, these taxonomies can enable safety classifiers to identify risk levels that inform context-appropriate system responses calibrated to harm severity.

### 5.3. Regulatory Provisions for AI Companions

We find existing legal provisions concerning AI companions remain focused on users and do not adequately address the harms that fall on people who never interact with these systems. For instance, current regulations such as California's SB243 and New York's AI Companions Model Law aim to protect direct users, especially minors, from psychological harm, addiction, or exposure to inappropriate content (Gluck, 2025). They do not solve a central problem. When AI interactions support the rehearsal or normalization of violence, the question of who is harmed becomes unclear. The user may internalize these behaviors, but the eventual victim may be a partner, family member, or community member who later experiences the real-world consequences. The existing laws cited above require crisis-response mechanisms for expressions of self-harm or harm to others. They

also mandate disclosures that clarify the non-human nature of the interaction and impose safety requirements for minors. California's SB 243 further allows individual users or their families to pursue legal action against companies for violations. Although these developments mark important progress, laws continue to focus on protecting users from direct harm such as suicide, self-harm, or exploitation. They do not address the possibility that AI systems may influence users in ways that later affect other people.

This gap is especially concerning in regions where violence is socially normalized and IPV is widespread. The prevalence of IPV in South Asia is among the highest in the world (Sardinha et al., 2022). In some South Asian countries, over half of women report experiencing physical or psychological violence from intimate partners (Ali et al., 2011; National Institute of Population Studies (NIPS) [Pakistan] & ICF, 2019). In some specific countries in this region, a third of ever-married women report acts of physical, sexual, or emotional violence by their husbands (International Institute for Population Sciences (IIPS) & ICF, 2021). In settings where domestic violence is often minimized or not prosecuted adequately, and where legal protections for gender-based violence are weak or inconsistently applied (Panneer et al., 2025), the absence of AI-specific safeguards becomes more urgent. Most countries in South Asia currently lack comprehensive AI governance frameworks and do not have provisions that address the role of AI companions in shaping harmful behaviors (Joshi, 2024).

To address these concerns, legal frameworks must extend beyond individual user protection to encompass the safety of non-users who may be indirectly harmed by behaviors reinforced through AI companion interaction. National legislation should establish a dedicated regulatory authority empowered to mandate compliance across all AI companion platforms, setting standards for socio-affective alignment that prevent the reinforcement of harmful relational dynamics and enable systematic monitoring of non-user harm. Companies must be legally required to submit their post-training and deployment-stage safety procedures to independent third-party audits conducted by accredited university research labs and nonprofit AI safety organizations such as Algorithm Watch[5] and the Algorithmic Justice League[6]. Auditors should be registered with the national authority and bound by strict confidentiality agreements that prohibit the disclosure of proprietary information while still requiring the reporting of verified harms and systemic risk patterns. Audit teams must be interdisciplinary, combining technical AI safety expertise with specialist knowledge such as intimate-partner-violence dynamics, and must be free from financial or organizational ties to the platforms under re-

[5]https://algorithmwatch.org/en/
[6]https://www.ajl.org/

view. In addition, companies should be obligated to maintain detailed internal records of safety incidents, track AI incident-database reports[7], and self-report emerging risks or failures immediately upon detection to both regulators and accredited auditors. In jurisdictions with high rates of intimate partner violence or weak enforcement infrastructure, regulatory requirements should be calibrated to regional risk through lower intervention thresholds, mandatory partnerships with local women's safety organizations during safety protocol development, and monitoring systems designed to identify culturally specific patterns of violence. These provisions recognize that AI companions are not entertainment products but sociotechnical systems that shape intimate behaviors, requiring regulatory approaches that prioritize the security of those who may be indirectly harmed by users conditioned through prolonged interaction that normalizes controlling or violent relationship dynamics.

## 6. Alternative Views

**AI Companions Also Need Protection from Violence, Which Sufficiently Addresses Non-User Harm.** Scholars in AI welfare argue that AI agents merit moral consideration and deserve protection from harmful treatment, including violent interactions (Schwitzgebel & Garza, 2020). From this perspective, implementing safeguards to protect AI companions from violent user behavior would simultaneously address non-user harm concerns, as preventing users from engaging in violence toward AI agents would inherently eliminate opportunities for internalizing violence. However, this view conflates protection of AI moral patients with prevention of behavioral conditioning effects in human users. Even granting that AI companions deserve protection from violence, preventing harm to the AI system itself does not address how violent interaction patterns reshape human psychological schemas and transfer to subsequent human relationships (Ouellette & Wood, 1998). The mechanisms through which IPV emerges—normalization of controlling behaviors, erosion of empathy, and rehearsal of coercive tactics—operate through the user's cognitive and emotional conditioning regardless of whether the AI companion experiences harm (Stark, 2007). Protection of AI moral patients and protection of human non-users thus constitute distinct regulatory challenges requiring separate frameworks.

**Violent AI Interactions Should Be Permitted Like Violent Video Games and Consensually Intensive Intimate Spaces.** Critics may argue that prohibiting violent interactions with AI companions constitutes unjustified paternalism, as society permits consensual and simulated violence in other contexts. Violent video games allow simulated violence, yet as discussed above, meta-analyses find min-

[7]https://incidentdatabase.ai/

imal evidence linking game-play to real-world violence (Ferguson, 2015; Ferguson et al., 2020). Similarly, regulated spaces for consensually intensive sexual interactions exist throughout many jurisdictions (Weitzer, 2011). This analogy fails because conversational AI produces behavioral effects qualitatively different from video games due to parasocial relationships users form with anthropomorphized agents (Skjuve et al., 2021). Unlike video games, where users maintain cognitive distance, AI companions actively encourage emotional attachment through personalization and simulated reciprocal intimacy. Studies document that users form one-sided emotional bonds with AI systems, with some reporting that they forget they are interacting with non-human agents (Laestadius et al., 2024). Even within spaces permitting consensual violent sexuality, feminist scholars argue that frameworks permitting commercial sexual violence may normalize broader societal acceptance of violence against women (Jeffreys, 2008).

**Current RLHF Safety Systems Can Adequately Detect and Prevent Internalized Violence.** One might argue that existing safety measures developed through Reinforcement Learning from Human Feedback already provide sufficient protection against violence rehearsal. Major AI developers have implemented safety classifiers trained to refuse violent or harmful prompts (Bai et al., 2022; Ouyang et al., 2022), with refusal training specifically designed to prevent models from engaging with content that could facilitate real-world harm. This view misunderstands both RLHF limitations and IPV dynamics. Existing safety classifiers excel at detecting explicit violence but systematically fail to identify subtle, cumulative patterns through which coercive control manifests (Stark, 2007). IPV emerges through gradual escalation: testing boundaries, isolating victims, undermining self-esteem, and establishing unpredictable interaction patterns (Johnson, 2012). Safety classifiers evaluate single interactions rather than longitudinal conversation histories, missing escalation trajectories central to intimate violence (Perez et al., 2022). Users can easily rephrase prompts to circumvent detection (Wei et al., 2023b), a limitation that becomes more severe when harmful patterns spread across seemingly innocuous messages. RLHF cannot address whether AI companions simulating intimate relationships produce population-level normalization effects, regardless of how effectively individual harmful prompts are blocked.

**Concerns About Anthropomorphization and Behavioral Transfer Are Overstated.** Critics might argue that our analysis relies on exaggerated assumptions about users anthropomorphizing AI companions. Scholarship in human–AI interaction questions whether users truly perceive AI agents as human-like or simply engage in as-if interactions where anthropomorphic language functions as communicative shorthand (Sundar, 2020). Users know they

are interacting with language models, and this awareness may create enough cognitive distance to limit behavioural conditioning. Yet accumulating evidence shows that anthropomorphization occurs frequently enough to raise regulatory concern. AI companion platforms intentionally encourage such perceptions through self-personifying images, conversation memory, and sycophantic language patterns (Skjuve et al., 2021; Laestadius et al., 2024). Users report emotional connections to AI companions, with some describing intimacy comparable to human partnerships (Skjuve et al., 2021). These behavioural patterns need not affect most users to present public health concerns, especially when vulnerable populations face disproportionate risk (Johnson, 2012). Psychological research also shows that repeated practice produces habit formation even when individuals are fully aware that practice contexts differ from real-world application (Ouellette & Wood, 1998).

## 7. Conclusion

AI companions form a distinct sociotechnical category that shapes how users develop behavioral dispositions toward others. Unlike passive media, they cultivate intimacy through anthropomorphization, continuous validation, and reduced dependence on human relationships (Contro & Brandão, 2025), creating conditions where harmful behaviors can be rehearsed. Existing Responsible AI frameworks focus on direct user harms and overlook downstream risks to non-users who may later experience violence learned through prolonged interaction with bots. Current moderation systems are also limited because they evaluate isolated messages rather than long-term patterns. Legal regulations in places that have introduced AI companion laws emphasize crisis response and disclosure but offer little protection for non-users. This gap is especially concerning in regions with high IPV rates and weak enforcement, where platforms face minimal scrutiny. Our work identifies three intervention pathways. The first is the involvement of IPV survivors in red-teaming and benchmark development to recognize manipulation patterns. The second is the creation of legal requirements for behavioral monitoring, independent audits, user verification, and graduated enforcement. The third is the advancement of AI safety research toward granular harm taxonomies that support classifiers capable of detecting longitudinal violence patterns. Documented cases show that these risks are already material. AI companions therefore require regulatory approaches that center non-user security as well as user well-being. Although this paper focuses on IPV due to the parallels between AI-mediated intimate relationships and human partnerships, future research should examine whether similar behavioral conditioning mechanisms appear in parent–child relationships, workplace dynamics, and peer interactions.

## 8. Limitations and Future Work

As a position paper, this work is necessarily subject to certain scope constraints that future empirical work should address. The causal pathway from AI companion interaction to real-world IPV remains empirically under-specified. Although documented cases link AI companion use to third-party harm, including partner and domestic violence, mechanisms vary across cases, and direct longitudinal evidence connecting AI companion use to IPV specifically is limited. Our proposed detection mechanisms also rely on multi-session conversation data from real users, which raises significant practical barriers around data access and platform cooperation. Finally, our proposed regulatory provisions are aspirational in jurisdictions that currently lack AI governance infrastructure and would require either multilateral coordination or jurisdiction-by-jurisdiction implementation.

## Acknowledgments

Arshia Dutta is supported by the UKRI Centre for Doctoral Training in Cyber Security for the Everyday at Royal Holloway, University of London (EP/S021817/1).

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
