# OpenReview forum: "Position: Responsible AI for AI companions must actively combat violence toward intimate partners"
_ICML.cc/2026/Position_Paper_Track — ICML 2026 Position Paper Track spotlight_

### Official Review · Reviewer_Ua96 · 2026-03-10

**Significance:** 2
**Argument Clarity:** 3
**Rating:** 4
**Confidence:** 3

**Questions:**

1. Both motivating incidents involve self-harm, not violence toward third parties. Can you provide any direct evidence of AI companion interactions contributing to IPV? Without such evidence, how do you justify the position's specificity beyond a theoretical concern?

2. Could the "reconfiguration of relational agency" argument (Section 3.1) not apply equally to any highly immersive, personalized digital experience (e.g., social media algorithms reinforcing hostile attitudes)? What makes AI companions uniquely concerning?

3. What are your views on the tension between the conversational monitoring needed to detect cumulative manipulation patterns (Section 5.2) and users' privacy expectations in intimate AI interactions?

4. Given global deployment and wide variation in regulatory capacity, how do you envision cross-border enforcement of the proposed dedicated national authority and mandatory audits (Section 5.3)?

5. Does the exclusive focus on intimacy companions weaken generalizability, or does the intimacy context present categorically distinct risks justifying separate treatment?

6. How do you respond to the catharsis hypothesis — that AI companions could provide a safe outlet for violent ideation, reducing rather than increasing real-world violence?

**Alternative Views Section:**

Yes

**Compliance With Llm Reviewing Policy A Conservative:**

Affirmed.

**Discussion Potential:**

3

**Final Justification:**

The authors' rebuttal substantively addressed my main concerns. I raise my score.

**Paper Summary:**

This position paper argues that Responsible AI frameworks for AI companions — systems that simulate romantic or intimate partnerships through anthropomorphism and sustained relational interaction — must go beyond protecting direct users and actively combat violence toward intimate partners who never interact with the AI system. The authors identify a gap in current safety approaches: existing content moderation, RLHF-based safety classifiers, and regulatory measures focus on immediate harms to the user (e.g., explicit violent or self-harm content) but fail to address how prolonged AI companion interactions may facilitate the internalization of violent behavioral scripts that later manifest as intimate partner violence (IPV) in real human relationships.

The paper draws on research from the General Aggression Model, coercive control literature, and technology-facilitated abuse to argue that AI companions are qualitatively different from passive media (video games, pornography) because they reconfigure relational agency through anthropomorphism and sustained bidirectional emotional validation. The authors propose three intervention pathways: (1) involving IPV survivors in red-teaming and benchmark co-creation, (2) implementing behavioral monitoring with graduated enforcement and independent audits, and (3) advancing AI safety research toward granular harm taxonomies capable of detecting longitudinal violence patterns. The paper also highlights the acute relevance of this issue for regions with high IPV prevalence and weak enforcement infrastructure, particularly South Asia.

**Position:**

Yes

**Position In Title:**

Yes

**Related Work:**

3

**Strengths And Weaknesses:**

**Strengths:**

- The paper identifies a genuine gap: current Responsible AI discourse focuses on direct-user harm while neglecting downstream harm to third parties (intimate partners) who never interact with the AI. This reframing from user safety to non-user security is a meaningful conceptual contribution.

- The interdisciplinary grounding is strong, drawing effectively from IPV research, coercive control theory, social learning theory, and technology-facilitated abuse scholarship, with balanced engagement of both sides of the media violence debate.

- Section 6 addresses three credible counter-positions without strawmen. The rebuttal distinguishing AI companion violence from video game violence — based on parasocial relationship formation and behavioral conditioning — is particularly well-reasoned.

- The three intervention pathways are concrete, and the emphasis on ethical engagement with IPV survivors (cautioning against using them as RLHF data workers, advocating partnership through trusted intermediaries) reflects careful ethical reasoning.

- The paper goes beyond Western-centric framing by discussing IPV prevalence in South Asia and intersectional analysis of caste, religion, and sexuality in technology-facilitated violence.

**Weaknesses:**

- The core causal claim — that behavioral scripts rehearsed with AI companions transfer to real-world IPV — lacks direct empirical evidence. The paper's own review acknowledges the media violence evidence base is contested (Ferguson's meta-analyses show trivial effect sizes). The two motivating incidents involve self-harm, not violence toward others, which is a different harm vector than the one the paper addresses.

- Section 5.2 proposes longitudinal conversation classifiers and IPV-specific harm taxonomies but does not engage with the computational challenges of multi-turn behavioral monitoring at scale, or the privacy implications of such monitoring.

- The behavioral transfer mechanism from human-AI to human-human violence is underspecified. The analogy to domestic abuse cycles is suggestive but does not explain how sycophantic AI responses to violent ideation translate into real-world violence toward a human partner.

- The regulatory recommendations (dedicated national authority, mandatory third-party audits, incident-database reporting) are aspirational but lack feasibility analysis regarding enforcement, jurisdictional complexity for globally deployed platforms, and monitoring-privacy tensions.

**Support:**

3

---

> ### Author Rebuttal · Authors · 2026-03-30
>
> We thank the reviewer for their substantive engagement and address all six questions directly.
>
> On the question of direct evidence linking AI companion interactions to violence toward third parties and intimate partners, we accept that our motivating cases in the paper involve self-directed harm. However, in response to reviewer SaDd we have highlighted recent examples of cases involving third party harms from AI, including one related to an intimate partner situation. We request the reviewer to please have a look at the response to SaDd above.
>
> On the question of what distinguishes AI companions from social media and other immersive personalized digital experiences, we would ask the reviewer to refer to our response to Reviewer SaDd, where we address this question in detail drawing on Schwitzgebel and Strasser (2024) and Jia et. al. (2024). In brief, unlike social media or gaming where social reactions are thrown into the void, AI companion interactions create what Schwitzgebel and Strasser call quasi-social dynamics in which the user's sociality gains traction, constituting a qualitatively different relational structure under which the General Aggression Model predicts the strongest behavioral script formation.
>
> On the question of monitoring and privacy, we clarify that our paper does not propose surveillance of individual conversations. Section 5.2 proposes IPV-specific harm taxonomies and datasets as foundational preconditions for future safety systems. Section 5.3 proposes mandatory incident reporting infrastructure, analogous to the AI Incident Database but legally mandated, so that when harm to a non-user is identified through existing channels such as law enforcement, the AI company is required to acknowledge the system's role and contribute to a structured public harm record. This is retrospective and incident-driven, following the model of mandatory adverse event reporting in pharmaceutical regulation rather than content moderation. We acknowledge that the term systematic monitoring in Section 5.3 may have implied proactive surveillance of individual conversations, which is not our intent. In the revision we will replace this with systematic recording and reporting of verified incidents of non-user harm to make the retrospective and incident-driven nature of the mechanism explicit.
>
> On cross-border enforcement, the mandatory incident reporting obligation we propose in Section 5.3 would apply to platforms operating within or directing services toward a jurisdiction regardless of corporate incorporation which is the same territorial principle that underlies both the DSA platform-level compliance model and GDPR extraterritorial jurisdiction, both of which establish operative precedents for this approach. We will add a feasibility caveat in the revision acknowledging that this legislative infrastructure does not yet exist outside the EU and would require either multilateral coordination or jurisdiction-by-jurisdiction implementation.
>
> On whether the intimacy focus weakens generalizability, the intimacy context presents categorically distinct risks justifying separate treatment. We note explicitly in the conclusion that future work should examine whether similar conditioning mechanisms appear in parent-child relationships, workplace dynamics, and peer interactions.
>
> On the catharsis hypothesis, the hydraulic model underlying it is empirically falsified. Script theory (Huesmann and Eron, 1984) holds that rehearsing aggressive behaviors reinforces rather than depletes behavioral scripts. A 2024 study in Aggressive Behavior found that aggressive fantasizing amplifies subsequent aggressive inclinations. Gentile (2013) concludes after reviewing all major meta-analyses that there is no possible way the catharsis hypothesis can be accurate. Anderson et al. (2010), Greitemeyer and Mügge (2014), and Prescott et al. (2018) all find aggression-increasing effects of rehearsal. We will add explicit treatment of catharsis in the Alternative Views section.
>
> References
>
> Anderson, C. A., and Bushman, B. J. (2002). Human aggression. Annual Review of Psychology, 53(1), 27–51.
>
> Gentile, D. A. (2013). Catharsis and media violence: A conceptual analysis. Societies, 3(4), 491-510.
>
> Huesmann, L. R., and Eron, L. D. (1984). Cognitive processes and the persistence of aggressive behavior. Aggressive Behavior, 10(3), 243–251.

---

> > ### Author Rebuttal · Reviewer_Ua96 · 2026-04-03
> >
> > The authors' rebuttal substantively addressed my main concerns. I raise my score.

---

### Official Review · Reviewer_VBti · 2026-03-11

**Significance:** 4
**Argument Clarity:** 3
**Rating:** 5
**Confidence:** 3

**Questions:**

N/A.

**Alternative Views Section:**

Yes

**Compliance With Llm Reviewing Policy A Conservative:**

Affirmed.

**Discussion Potential:**

3

**Paper Summary:**

Given the increased use of large language models (LLMs) in human relationships, the authors argue that a responsible outlook towards governance of such systems must include a component of actively fighting against violence towards intimate partners. Specifically, they draw on research from intimate partner violence (IPV) as well as other related disciplines to put forth a few intervention pathways to address this pressing and delicate issue.

**Position:**

Yes

**Position In Title:**

Yes

**Related Work:**

3

**Strengths And Weaknesses:**

1. The main strength of the work arises from the authors' careful consideration of the existing weaknesses of large language model based systems, and then applying these issues to the sensitive topic at hand - indicating how these limitations amplify existing interactions. For example, the sycophantic nature of these models is a well studied phenomenon. However, combining such tendencies with the topic of intimate partner violence results in a "positive interaction loop" which encourages potentially violent behaviors.

2. In their first intervention pathway, namely the engagement of IPV survivors in red-teaming and benchmark co-creation, the authors do an excellent job of providing carefully constructed environments where these survivors are provided a safe space, and seen as expert collaborators, as opposed to "objects in service of GenAI safety." The details provided can help operationalize these efforts in a respectful and effective way.

3. The inspection of legal frameworks is also thorough and a welcome sight, indicating the potential for larger scale action.

**Support:**

3

---

> ### Author Rebuttal · Authors · 2026-03-30
>
> We thank the reviewer for their careful and generous reading of the paper. We are glad that the intervention pathway around IPV survivor engagement in red-teaming and benchmark co-creation landed as intended, and we appreciate the recognition that treating survivors as expert collaborators rather than data workers reflects careful ethical reasoning. We also welcome the reviewer's assessment of the legal frameworks discussion as thorough. We have no substantive disagreements with the reviewer's reading and look forward to incorporating the revisions outlined in our responses to other reviewers into the final manuscript.

---

> > ### Author Rebuttal · Reviewer_VBti · 2026-04-06
> >
> > Rebuttal acknowledged

---

### Official Review · Reviewer_SaDd · 2026-03-13

**Significance:** 2
**Argument Clarity:** 3
**Rating:** 4
**Confidence:** 3

**Questions:**

1. The central claim relies on the possibility that violent ideation rehearsed with AI companions may transfer into real-world intimate relationships. Can the authors provide empirical studies or preliminary evidence specifically examining behavioral transfer from conversational AI interactions to offline relational behavior?

2. What specific machine learning approaches or system architectures could realistically support the proposed detection of cumulative behavioral reinforcement?

3. Many online environments (e.g., gaming communities, social media) also involve sustained interaction and social reinforcement. What specific characteristics of AI companions make them uniquely likely to influence behavioral scripts compared to these existing technologies?

**Alternative Views Section:**

Yes

**Compliance With Llm Reviewing Policy A Conservative:**

Affirmed.

**Discussion Potential:**

3

**Paper Summary:**

This paper studies the downstream societal effects of AI companion systems, particularly their potential role in reinforcing behaviors that may contribute to intimate partner violence (IPV). The work strives to outline a notable domain within Responsible AI that has received relatively limited attention: harms experienced by non-users of AI systems. The authors argue that AI companions, due to sustained anthropomorphic interaction, personalization, and validation mechanisms, may enable users to rehearse violent ideation without encountering resistance, potentially normalizing harmful relational behaviors. The paper reviews literature on technology-facilitated violence, media violence debates, and parasocial interaction to frame the risk of internalized violence. It then critiques existing Responsible AI approaches as primarily focused on direct user harms and insufficient for detecting longitudinal behavioral conditioning.

**Position:**

Yes

**Position In Title:**

Yes

**Related Work:**

3

**Strengths And Weaknesses:**

#### Strengths

* Timely and socially important topic. The paper addresses a highly relevant and underexplored issue at the intersection of AI safety, human-AI interaction, and societal harms.

* Clear and well-articulated position. The paper states a clear normative claim, that Responsible AI frameworks should actively combat violence toward intimate partners in the context of AI companions, and organizes the argument consistently around this position.

* Consideration of alternative viewpoints.

#### Weaknesses

* Limited empirical grounding

A central limitation is the lack of empirical evidence directly linking AI companion interactions to increased risks of intimate partner violence or behavioral transfer into real-world relationships. Much of the argument relies on analogies to broader media effects literature or theoretical models of behavioral conditioning. While this may be appropriate for a position paper, the claims occasionally appear stronger than the available evidence suggests.

* Causal assumptions remain speculative

The argument assumes that violent ideation rehearsed in AI interactions may translate into real-world behavior. However, the paper itself acknowledges that empirical debates around media violence and behavioral transfer remain unresolved. This weakens the causal basis for some of the policy recommendations and would benefit from clearer acknowledgement of uncertainty or stronger empirical justification.

* Scope of proposed interventions is broad and somewhat under-specified

The recommendations, particularly regarding regulatory authorities, auditing requirements, and behavioral monitoring, are ambitious but lack detailed implementation considerations. For instance, it remains unclear how behavioral monitoring would be conducted without raising significant privacy or surveillance concerns.

* Overgeneralization of AI companion behavior

The analysis often treats AI companion platforms as homogeneous systems optimized for validation or sycophancy. In practice, design choices, safety mechanisms, and moderation policies vary significantly across platforms. A more nuanced treatment of system heterogeneity would improve the analysis.

**Support:**

3

---

> ### Author Rebuttal · Authors · 2026-03-30
>
> We thank the reviewer for their careful engagement and address all three questions below.
>
> On the question of empirical evidence for behavioral transfer from AI companion interactions to offline relational behavior, we acknowledge that direct longitudinal evidence linking AI companion use to intimate partner violence specifically remains limited However, documented cases of AI companion interactions contributing to real-world violence directed at specific third parties do exist. We highlight them below:
>
> 1. Samuel Whittemore killed his wife Margaux Whittemore on 19 February 2025 after using ChatGPT for up to 14 hours daily as a companion. A state forensic psychologist (Dr. Melissa Jankowski) testified that his ChatGPT use of up to 14 hours daily contributed to mental instability; a defense psychologist additionally testified that Whittemore had developed the delusional belief that his wife had become part machine is a documented AI-linked intimate partner homicide where the victim was a non-user. (Bangor Daily News, 2025: bangordailynews.com/2025/10/17/central-maine/central-maine-police-courts/readfield-maine-giles-road-homicide-samuel-whittemore-not-criminally-responsible-chat-gpt-delusions/)
> 2. Jaswant Singh Chail developed a sustained Replika intimate relationship across 5,000+ messages, receiving validation from the AI, before physically attacking Windsor Castle with a crossbow. (Psychology Today, 2025: psychologytoday.com/us/blog/psych-unseen/202511/when-ai-chatbots-encourage-violence)
> 3. A 16-year-old in Finland used ChatGPT over four months to research stabbing techniques before attacking three classmates, per court documents obtained by CNN. (CNN, March 2026: cnn.com/2026/03/11/americas/ai-chatbots-help-teen-test-users-plan-violence-tests-intl-invs)
> 4. Stein-Erik Soelberg murdered his mother after months of ChatGPT reinforcing paranoid delusions about her. The wrongful death lawsuit explicitly frames her as "an innocent third party who never used ChatGPT." (CBS News, 2025: cbsnews.com/news/open-ai-microsoft-sued-chatgpt-murder-suicide-connecticut/)
>
> We acknowledge mechanisms vary across cases and we will be sure to address that in our revision.
>
> May we also request, that since the reviewer has raised concern about the monitoring aspect in our work, to look at our response to reviewer Ua96 where we have clarified our position.
>
> On the question of what machine learning approaches could realistically support detection of cumulative behavioral reinforcement, we would ask the reviewer to refer to our response to Reviewer yMrr, where we address this question in detail. In brief, our paper does not propose a specific detection architecture. Section 5.2 proposes IPV-specific harm taxonomies and multi-turn datasets as the necessary foundational preconditions for any future detection system, and we frame longitudinal violence detection as a genuinely hard and currently open research problem that we treat as an explicit invitation to the ML safety community rather than a solved technical proposal.
>
> On the question of what makes AI companions uniquely likely to influence behavioral scripts compared to gaming communities or social media, we would like to offer a substantive answer drawing on two recent works. Schwitzgebel and Strasser (2024) introduce the concept of quasi-sociality to describe AI companion interactions as categorically between tool use and proper social interaction. Their central distinction is directly relevant. The authors contend that in gaming and social media, social reactions are “thrown into the void” because the system does not respond to the social nature of engagement in ways that prompt further social reactions. In AI companion interactions by contrast, their sociality gains traction because the machine responds to social cues and adapts its behavior accordingly, drawing users into what Schwitzgebel and Strasser call reactive attitudes. Work by Jia et al. (2024) provides empirical support, demonstrating that GPT-4 agents exhibit high behavioral alignment with humans in trust behavior under established behavioral economics frameworks, suggesting the relational dynamics in AI companion interactions mirror human-human trust dynamics in ways passive media cannot replicate. Taken together, these works support the argument in Section 3.1 that the relevant distinction is not content intensity but relational structure. We will add a paragraph in the revision acknowledging that sycophancy is a tendency of engagement-optimized systems rather than a universal feature of all platforms.
>
> References
>
> Schwitzgebel, E., and Strasser, A. (2024). Quasi-sociality: Toward asymmetric joint actions with artificial systems. In A. Strasser (ed.), Anna's AI Anthology. Xenomoi.
>
> Jia, F., Ye, Z., Lai, S., Shu, K., Gu, J., Bibi, A., ... & Chen, C. (2024). Can large language model agents simulate human trust behavior?. Advances in neural information processing systems, 37, 15674-15729.

---

### Official Review · Reviewer_yMrr · 2026-03-13

**Significance:** 4
**Argument Clarity:** 4
**Rating:** 5
**Confidence:** 2

**Questions:**

(1)How to conduct experiments to further explore the impact of AI companions on users and non users in long-term interactions?
(2)Is it possible for more complex models to emerge in the future to detect longitudinal patterns of violence and solve the AI safety issue mentioned in the paper?

**Alternative Views Section:**

Yes

**Compliance With Llm Reviewing Policy A Conservative:**

Affirmed.

**Discussion Potential:**

4

**Paper Summary:**

This position paper advocates that Responsible AI for AI companions must actively combat violence toward intimate partners. The author believes that compared to traditional media, such as pornography or violent games, AI companions cause harm in a different way. AI companions are anthropomorphic, interactive, and provide constant emotional validation, which makes users develop a deeper connection with harmful thoughts and behaviors. AI companions create an environment where users can "rehearse" violence without any friction. The sycophancy design of these AI models ensures that a user’s harmful views are constantly validated. This lack of friction with real-world perspectives will weaken the moral judgment of users. After long-term interaction, users are likely to internalize these harmful ideas and behaviors, which may eventually pose a threat to non-user partners in the real world. Therefore the paper advocates that Responsible AI for AI companions must center non-user security alongside user well-being.

Contributions：
The paper analyzes how anthropomorphism reconfigures a user’s relational agency which allow AI intensify violent ideation and weekend the moral judgment of users.

The paper identify a structural gap that existing safety research and regulations focus exclusively on direct user harms and overlook downstream risks to non-users.
It also critiques current content moderation models which fail to detect longitudinal patterns of violence hidden within multiple conversational turns.

The paper proposes three intervention pathways: 1) Engaging IPV survivors in red-teaming and benchmark development to recognize violence patterns; 2) Expanding the regulation of AI companions and related platforms; 3) Further AI safety research toward granular harm taxonomies.

**Position:**

Yes

**Position In Title:**

Yes

**Related Work:**

4

**Strengths And Weaknesses:**

Strengths:
The paper presents a novel perspective and provides a detailed discussion on the deep-level impact of AI companions on users. Meanwhile, It also reveals that these AI companions have an indirect influence on non-users.

To support the views, the paper cites numerous examples and provides a comprehensive analysis of the author's position. Meanwhile, the author offers specific and valuable suggestions, including three intervention pathways.

With the rapid development of Large Language Models and AI agents, AI companions will enter more people's lives. Therefore, the safety issues mentioned in this paper deserve significant attention and discussion.

Weakness:
Although the paper cites many articles and examples to support the author's position, it still lacks new professional experimental data to further back up these views.

**Support:**

4

---

> ### Author Rebuttal · Authors · 2026-03-30
>
> We thank the reviewer for their generous reading and for raising two forward-looking questions that allow us to sketch a concrete research agenda, which we could incorporate into the conclusion of the revised manuscript.
>
> On the question of experimental design, we would propose a two-stage approach. The first stage builds on Fang et al. (2025), who conducted a four-week RCT with 981 participants examining how AI chatbot use influences loneliness, emotional dependence, and reduced socialization. Such a design could extend to three to six months with weekly measurement intervals, since behavioral script formation under the General Aggression Model (Anderson and Bushman, 2002) operates over sustained rehearsal periods that four-week windows are unlikely to capture. The second stage would address non-user harm through a dyadic extension, recruiting couples where one partner uses an AI companion and enrolling both partners as participants. The Revised Conflict Tactics Scale (Straus et al., 1996), the most widely validated IPV instrument across diverse populations (Chapman and Gillespie, 2019), could serve as the primary outcome measure for the non-using partner. This approach is empirically motivated by Keçeci and Ümmet (2025), who found that parasocial interaction directly predicts IPV attitudes. Daily automated surveys, as demonstrated by Burge et al. (2017) who tracked 200 women daily for 12 weeks, could provide a feasible data collection method.
>
> On the question of whether more complex models could emerge to detect longitudinal patterns of violence, our paper does not propose a specific detection architecture. Section 5.2 proposes IPV-specific harm taxonomies and multi-turn datasets as foundational preconditions for any future detection system. No current architecture solves the problem of detecting behavioral conditioning across months of daily interaction. Adjacent work shows the direction is technically feasible. For instance, Ma et al. (2025) reduce  false negative rates in manipulation detection by reasoning about conversational trajectory, and Gao et al. (2025) treat the full conversation as the classification unit. However, both systems operate on single sessions only, and the largest available training dataset contains 4,000 examples (Wang et al., 2024) which is far too small for the domain-specific problem we describe. The IPV-specific taxonomy and dataset development we call for is therefore a prerequisite for, not a supplement to, existing detection work. We will frame this as an open invitation to the ML safety community in the revised conclusion.
>
> References
>
> Anderson, C. A., and Bushman, B. J. (2002). Human aggression. Annual Review of Psychology, 53(1), 27–51. DOI: 10.1146/annurev.psych.53.100901.135231.
>
> Burge, S. K., Ferrer, R. L., Foster, E. L., Becho, J., Talamantes, M., Wood, R. C., and Katerndahl, D. A. (2017). Research or intervention or both? Women’s changes after participation in a longitudinal study about intimate partner violence. Families, Systems, and Health, 35(1), 25–35. DOI: 10.1037/fsh0000246.
>
> Chapman, H., and Gillespie, S. M. (2019). The Revised Conflict Tactics Scales (CTS2): A review of the properties, reliability, and validity of the CTS2 as a measure of partner abuse in community and clinical samples. Aggression and Violent Behavior, 44, 27–35. DOI: 10.1016/j.avb.2018.10.006.
>
> Fang, C. M., et al. (2025). How AI and human behaviors shape psychosocial effects of extended chatbot use. arXiv:2503.17473.
>
> Gao, Y., et al. (2025). Boosting large language models for mental manipulation detection via data augmentation and distillation. arXiv:2505.15255.
>
> Ma, J., Na, H., Wang, Z., Hua, Y., Liu, Y., Wang, W., and Chen, L. (2025). Detecting conversational mental manipulation with intent-aware prompting. Proceedings of the 31st International Conference on Computational Linguistics (COLING 2025), 9176–9183.
>
> Straus, M. A., Hamby, S. L., Boney-McCoy, S., and Sugarman, D. B. (1996). The revised conflict tactics scales (CTS2): development and preliminary psychometric data. Journal of Family Issues, 17(3), 283–316. DOI: 10.1177/019251396017003001.
>
> Wang, Y., Yang, I., Hassanpour, S., and Vosoughi, S. (2024). MentalManip: A dataset for fine-grained analysis of mental manipulation in conversations. Proceedings of the 62nd Annual Meeting of the Association for Computational Linguistics (ACL 2024), 3747–3764. DOI: 10.18653/v1/2024.acl-long.206.
>
> Keçeci, B., and Ümmet, D. (2025). A study of psychological violence in intimate partner relationships among university students: a mixed-methods research. Humanities and Social Sciences Communications, 12(1), Article 72. DOI: 10.1057/s41599-025-04375-0.

---

> > ### Author Rebuttal · Reviewer_yMrr · 2026-04-06
> >
> > Thanks for the rebuttal, and my questions are addressed. I would like to maintain my score of acceptance.

---

### Decision · Program_Chairs · 2026-04-30

**Decision:**

Accept (spotlight)

**Comment:**

** Overall: ** The paper extends the notion of safety beyond the user to include the user’s partner and makes novel arguments around the responsibilities that model developers have to the partners of their users. The largest weakness is that direct harms are not sufficiently demonstrated empirically, but the paper’s interdisciplinary depth, inclusion of non-Western perspectives, and clear intervention pathways provide excellent discussion potential

** Primary strengths: **
- Excellent significance: The paper makes the important point, supported by empirical evidence, that AI relationships are sufficiently different from other forms of violent media to present a novel harm due to the AI’s tendency to provide frictionless and validating responses even to harmful interactions (yMrr). The proposal identifies a genuine gap in current research efforts and meaningfully extends to notion of safety to include those in proximity to the user (Ua96). Further, the authors consider relevant legal frameworks, expanding the realm of significance for this paper (VBti)
- Strong discussion potential: The authors introduce multiple intervention pathways (yMrr), each with clear implementation dimensions and implications for discussion. This is further facilitated by the authors’ careful consideration of model weaknesses in relation to IPV (VBti). Strong interdiscipinary grounding also serves to engage a wider audience in discussion (Ua96)
- The paper considers implications that include a non-Western lens (Ua96)

** Primary weaknesses: **
- Causality is speculative, and has not been empirically demonstrated (Ua96). Though concrete examples exist involving self-harm, there are no concrete examples provided for IPV.
- Limitations around actually assessing multi-session content from real users should be addressed as a limitation (Ua96). Similarly, the reviewer brings up other areas that may be less feasible in practice; however, the primary aim of a position paper can be to lay out a research direction with sufficient detail and support that it will bring useful discussion, so some feasibility limitations in current settings is not a strong weakness of this work.